# Predicting Antimicrobial MICs for Nontyphoidal Salmonella Using Multitask Representations Learning of Transformer

## Abstract

The antimicrobial resistance (AMR) pathogens have become an increasingly worldwide issue, posing a significant threat to global public health. To obtain an optimized therapeutic effect, the antibiotic sensitivity is usually evaluated in a clinical setting, whereas traditional culture-dependent antimicrobial sensitivity tests are labor-intensive and relatively slow. Rapid methods can greatly optimize antimicrobial therapeutic strategies and improve patient outcomes by reducing the time it takes to test antibiotic sensitivity. The booming development of sequencing technology and machine learning techniques provide promising alternative approaches for antimicrobial resistance prediction based on sequencing. In this study, we used a lightweight Multitask Learning Transformer to predict the MIC of 14 antibiotics for Salmonella strains based on the genomic information, including point mutations, pan-genome structure, and the profile of antibiotic resistance genes from 5,278 publicly available whole genomes of nontyphoidal Salmonella. And we got better prediction results (improved more than 10% for raw accuracy and 3% for accuracy within ±1 2-fold dilution step) and provided better interpretability than the other ML models. Besides the potential clinical application, our models would cast light on mechanistic understanding of key genetic regions influencing AMR.

## 1 Introduction

Antibiotics are chemical compounds that are used for killing or inhibiting the growth of bacteria, playing a pivotal role in the control of infectious diseases. However, the ever-increasing antimicrobial resistance (AMR) threatens the clinical effectiveness of antibiotic treatments. The antibiotic resistance of pathogens could result in treatment failure, including high morbidity or mortality, and increase the health care cost substantially. Over 70 percent of the bacteria which promote hospital-acquired infections are resistant to at least one common antibiotic used for treatment (Stone et al., 2009).

In clinical settings, testing the antimicrobial resistance of pathogens is critical for the appropriate choice of antibiotics in the treatment. Antimicrobial susceptibility/ sensitivity testing (AST) is an approach to determine whether antibiotics can inhibit the bacteria/fungi growth, thus measure the susceptibility, or reflect the resistance of bacteria/fungi to specific the antibiotics. Several AST methods are widely used, including broth microdilution, antimicrobial gradient, disk diffusion test, and rapid automated instrument methods (Barth et al., 2009). Minimum inhibitory concentration (MIC) is one of the most frequently used AST methods, quantifying the lowest concentration of antibiotics preventing the growth of a microorganism. Qualitative descriptions (resistant/sensitive, etc.) of the antimicrobial sensitivity provide no accurate quantification of antimicrobial sensitivity and limit its power in certain scientific and clinical applications. In contrast, MIC measures provide a competent resolution while antimicrobial susceptibility of strains varies in a population, and this is useful for many epidemiological and clinical objectives.

Since traditional antimicrobial sensitivity testing relies on culture-dependent methods, it is labor-intensive and relatively slow. In the conventional microbiological laboratory diagnosis, the total time for the bacteria growth, isolation, taxonomic identification, and antibacterial MIC determination for

fast growing bacteria may exceed 36h, while the time for slowly growing bacteria may be several days (Opota et al., 2015). From a clinical point of view, testing the antimicrobial sensitivity using more accurate and rapid methods could greatly optimize antimicrobial therapeutic strategies and improves patient outcomes (Llor et al., 2014).

Whole-genome sequencing (WGS) has been widely used for public health surveillance in the past decades, guiding the clinical diagnosis and health care decisions. WGS-based data mining assesses the phylogenetic relationships, conducts outbreak investigations, detects antimicrobial resistance, and predicts the virulence or pathogenicity of potential pathogens (Varma et al., 2002). Several recent studies have used WGS data to predict AMR phenotypes. The most common approach relies on the homology search in a reference set of antimicrobial resistance genes and polymorphisms associated with them (Stoesser et al., 2013). This reference-guided and homology search approaches could describe antimicrobial resistance in a rough way if the targeted organisms have been adequately studied and the mechanisms of antimicrobial resistance are known. But the demand for more accurate and quantitative prediction of the antibiotic sensitivity or resistance necessitates novel predictive models.

With the increase of publicly available full-genome sequences, machine learning modelling have been developed to predict the antimicrobial sensitivity based on WGS data in recent studies. Some advanced statistical or machine learning (ML) models, including logistic regression (LR), gradient boosted decision trees (GBoost), Random Forests (RF) and deep neural networks (DNN), have been applied in predicting the antimicrobial sensitivity (Bálint., 2016). Based on the whole genome sequences of different strains and the corresponding MIC information, the predictive models could identify critical genes or regions associated with antimicrobial resistance without a priori information (Zankari, 2012). One study adopted 4 machine learning methods, including Random Forest, Gradient boosted decision trees, Deep neural networks, and Rule-based baseline to analyze whole-genome sequencing data of E. coli and predict antibiotic resistance. Using the presence or absence of genes, population structure, and year of isolation as predictors. Without prior knowledge of the causal mechanism, the Gradient boosted decision trees model achieved an average accuracy of 0.92 and a recall rate of 0.83 (Moradigaravand , 2018). Another study analyzed 704 E. coli genomes by using MIC measurements for ciprofloxacin. The models identified that 3 mutations in gyrA, 1 mutation in parC and presence of any qnrS gene, collectively associate with the MIC strongly (Kouchaki , 2018). Although such predictive approaches require many genomes and experimentally validated MIC for modelling, they are unbiased, accurate and able to discover genomic features associated with the AMR.

Salmonella is one of the most common causes of foodborne diseases, including stomach flu (gastroenteritis) and diarrhea, in the world, causing about 80 million illnesses all over the word annually (World Health Organization, 2015). Among Salmonella isolates, antimicrobial resistance is widespread, and infections caused by antibiotic-resistant strains are worse than those caused by antibiotic-susceptible strains (Varma, 2005). As a result of surveillance efforts by public health agencies, many whole-genome sequences and antimicrobial susceptibility data of Salmonella strains have been obtained (Hunt et al., 2017). One recent study adopted machine learning model called extreme gradient boosting (XGBoost) to predict MICs of 15 antibiotics based on over five thousand nontyphoidal Salmonella genomes (Marcus et al., 2017). The overall average accuracy of this MIC prediction models is 95% within ±1 2-fold dilution. Despite the excellent predictive performance of the model, the k-mers (features used) identified with highest contribution to the model offer weak biological interpretability. To understand how different genomic features, contribute to the antimicrobial resistance, machine models with the more interpretable features, including the copy number of antibiotic resistance genes and particular polymorphisms, etc., instead of k-mers, should be developed. The transformer model (based on the paper Attention is All You Need) follows the same general pattern as a standard sequence to sequence with attention model. The input sentence is passed through N encoder layers that generates an output for each token in the sequence. The decoder attends on the encoder's output and its own input (self-attention) to predict the next word. The transformer model has been proved to be superior in quality for many sequence-to-sequence problems while being more parallelizable. Here, we are going to do genome analysis and MIC prediction that is not sequence-to-sequence problem. So, only use transformer encoder. Self-attention network (SANs) can capture long-distance dependencies by explicitly attending to all the elements, regardless of distance. However, multiple relatively distant parts of a long genome sequence can work together, which may be overlooked by single attention. To solve this problem, we use the

multi-head attention mechanism. Enable its many aspects to process a sequence element at the same time to depict global characteristics. In addition, using multiple heads of attention plus a full connection layer can make training inexpensive and effective. Multi-task learning is an inductive transfer method that uses domain information contained in relevant task training signals as inductive bias to improve generalization ability. Multitask learning is prevalent in several applications, including computer vision, natural language processing (Augenstein et al., 2018), speech processing (Wu et al., 2018), and reinforcement learning (Rusu et al., 2016). It is achieved by using shared representation for parallel learning tasks; What we learn from each task can help us learn the other tasks better. This method can be well used in MIC prediction problems of various antibiotics to improve the generalization ability of the model and save training time. We tried some methods to optimize the weight in the multi-task learning. We adjust task-level loss coefficients dynamically by calculating task prioritization at both example-level and task-level. We also used Curriculum learning mode to facilitate the specific sub-tasks with poor performance by initializing the training process with easy tasks. All these measures improve the prediction accuracy of the model. In this study, we will adopt the pan-genome information, profile of antimicrobial resistance genes, single nucleotide variants (SNVs) in the resistance genes based on 5,278 publicly available whole genomes of nontyphoidal Salmonella with MIC information for 14 antibiotics. We first adopt a lightweight Multitask Learning Transformer to identify important features according to their statistical association with the MIC and generate a set of more interpretable features. Our model could be potentially used to guide antibiotic stewardship decisions for the nontyphoidal Salmonella.

## 2 Research Methods and Preliminary Data

### 2.1 Data used

A total of 5,278 genome sequences of nontyphoidal Salmonella were achieved from NCBI. All data was collected and sequenced as part of the NARMS program (Tay et al., 2019). We collected another 89 Nontyphoidal Salmonella reference genomes (Tay et al., 2019) to construct the Salmonella pan-genome database with MIDAS.

### 2.2 Identification of antimicrobial resistance genes, pan-genome information, and single nucleotide variants (SNVs)

Protein sequences of 89 reference genomes were annotated with Resfams (Heaton et al., 2017) to generate reference antibiotic resistance genes for Salmonella. Based on the pan-genome database built with MIDAS and the antibiotic resistance genes annotated, we estimate the copy number of each gene and single nucleotide variants (SNV) for each strain using MIDAS. The copy number of each gene, including the antibiotic resistance genes, and the SNV information will be used as raw features for downstream analysis. The number of single nucleotide alleles for one strain is 180000+.

### 2.3 Predictive modelling

#### 2.3.1 Feature selections

To reduce the dimensionality and complexity of the model, we used XGBoost Model filter to select features preliminarily. This model can rank the input features based on how deterministic they are for the laboratory-derived MICs in our data set. We currently evaluated features of the SNVs of antibiotic resistance geneto find out the important features before feed the data into the prediction model. (Fig. 1) XGboost, however, achieved an average prediction accuracy within ±1 2-fold dilution step of 90%. This is a reliable result, but not very accurate.

#### 2.3.2 Transformer Predictive Model

With the significant success of deep neural networks in the computer version, speech recognition and natural language processing (NLP), many deep learning-based models were proposed to solve biomedical problems. In this work, we will focus on sequence representation learning approaches. We propose a lightweight transformer model to explore the correspondence between the genome sequence and MIC of each antibiotic for each strain. The Fig. 2 is the architecture of our model.

The original sequences were put into XGboost filter module at first, after filter, the input sequences have been shrunken to 2000(1000, 500, 200, 100). Then they are feed to Encoder layers. Our transformer has 3 Encoder Blocks. We used 2 heads of attention in each Encoder Blocks. Full connection layers are used for MIC prediction task of 14 antibiotics. Each full connection layer corresponds to a task.

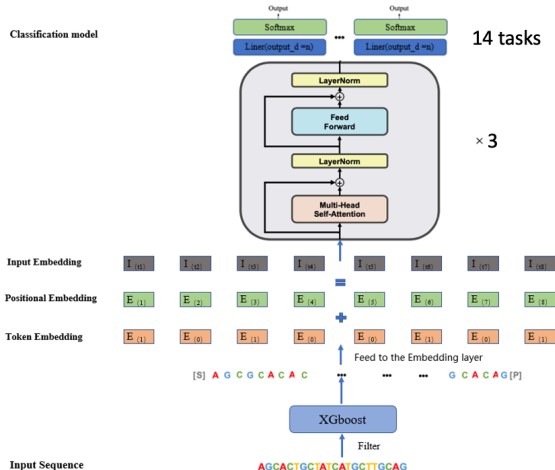

Figure 1: Architecture of our Transformer model

### 2.3.3 MULTI-TASK LEARNING

Multi-task learning is an inductive transfer method that uses domain information contained in relevant task training signals as inductive bias to improve generalization ability. It does this by learning tasks in parallel while using a shared representation. The purpose of sharing representation is to improve generalization. There are two ways for sharing representation of multiple tasks in shallow sharing parameter MTL: Parameter based sharing, such as neural network based MTL and Gaussian processing. Regularization based sharing, such as mean, Joint Feature learning (create a common feature set) Multi-task is usually believed to improve network performance as multiple related tasks help regularize each other and a more robust representation can be learned. In addition, combining all tasks into the same model also helps reduce computation. In our project, we have 14 tasks, after XGboost feature filter, we build a common feature set for these tasks. These tasks share the model parameters and the attention weight matrix.

### 2.3.4 DYNAMIC TASK PRIORITIZATION

Multitask learning models are sensitive to task weights (Augenstein et al., 2018). A task weight is commonly defined as the mixing or scaling coefficient used to combine multiple loss objectives. Task weights are typically selected through extensive hyperparameter tuning. However, there are already some methods attempt to dynamically adjust or normalize the task weights according to prescribed criteria or normalization requirements, such GradNorm (Fernando et al., 2016). These dynamic techniques are referred to as self-paced learning methods. In our project, we used learning progress signals to automatically compute a time-varying distribution of task weights. We dynamically adjust task-level loss coefficients to continually prioritize difficult tasks. Our loss uses learning progress signals to automatically calculate sample level and task level priorities. Our loss is:

$$\mathcal{L}_{\text{Total}}(\cdot) = \sum_{t=1}^{|T|} \mathsf{W} \mathcal{L}_t * \mathcal{L}_f(\cdot)$$

The first loss is the Example-level cross entropy loss with classes weight, the second loss is the Task-level cross entropy loss. Our experience shows that using Dynamic Task Prioritization can achieve

Table 1: Results of our models compared with the baseline

| Antibiotic | Raw Accuracy | baseline | Accuracy within ±1 2-fold dilution step | baseline | strains |
|---|---|---|---|---|---|
| All | 0.70 | 0.59 | 0.98 | 0.95 | 69112 |
| AMP | 0.71 | 0.33 | 1.00 | 0.92 | 5278 |
| AUG | 0.69 | 0.48 | 0.99 | 0.93 | 5278 |
| AXO | 0.87 | 0.8 | 0.96 | 0.95 | 5278 |
| CHL | 0.75 | 0.72 | 1.00 | 0.99 | 5278 |
| CIP | 0.64 | 0.42 | 0.99 | 0.97 | 5278 |
| COT | 0.78 | 0.87 | 0.99 | 0.98 | 5278 |
| FOX | 0.66 | 0.58 | 0.95 | 0.96 | 5278 |
| GEN | 0.52 | 0.46 | 0.90 | 0.91 | 5278 |
| NAL | 0.77 | 0.62 | 1.00 | 0.96 | 5278 |
| TET | 0.85 | 0.47 | 1.00 | 0.9 | 5278 |
| TIO | 0.67 | 0.73 | 1.00 | 0.99 | 5278 |
| AZI | 0.67 | 0.58 | 1.00 | 0.97 | 2415 |
| FIS | 0.54 | 0.57 | 0.97 | 0.95 | 4926 |
| STR | 0.61 | 0.51 | 0.97 | 0.93 | 2790 |

competitive performance. Using learning progress signals to automatically adjust the weight of tasks is useful for muti-task learning.

## 3 EXPERIENCE RESULTS

The goal is to dynamically prioritize difficult tasks during multitask learning. The main objective of this project is to predict MIC of 14 antibiotics for Salmonella with Transformer models based on whole genomic features and the corresponding MIC information of 5,278 nontyphoidal Salmonella genomes and identify key MIC-associated genomic features, including copy number and polymorphism of genes or motifs, etc., which contribute predominantly to our predictive model.

### 3.1 RAW ACCURACY AND ACCURACY WITHIN ±1 2-FOLD DILUTION STEP FOR MIC PREDICTION

Our prediction models had an overall average accuracy of 98% within ±1 2-fold dilution step, which improved 3% than baseline. For raw accuracy, we improved 10.1% than baseline. The result is showed in Table 1

### 3.2 VISUALIZATION FOR TRANSFORMER

More recently, Transformers have become a leading tool in deep learning application. The importance of Transformer networks necessitates tools for the visualization of their decision process. Visualizer is a small tool to assist the visualization of Attention module in the deep learning model. The main function is to help retrieve the Attention Map nested in the depth of the model. It is fast and convenient, and simultaneously pulls out all the Attention Maps in the Transformer class model. We use this tool to visualize the attention weight in our transformer. The Figure 2 is all attention weight result of one layer. There are 2 heads in each of our Encoder Block.

Then we can calculate the attention values for the features, and Figure 3 is the visualization of attention values for 500 features. We can see from the figure that the first head didn't catch valid information from the input features, but for the second head, the effective information is obtained from the sequence, and finally leads to a good prediction of the model. This is an intuitive and simple example of the effectiveness of the multi-head attention mechanism.

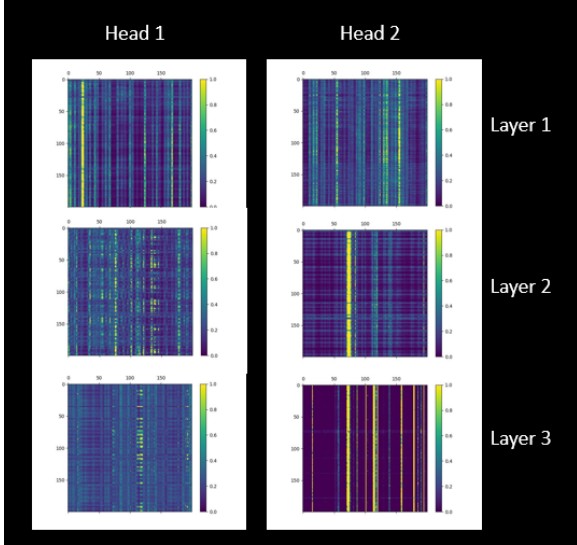

Figure 2: Visualization for Attention Map

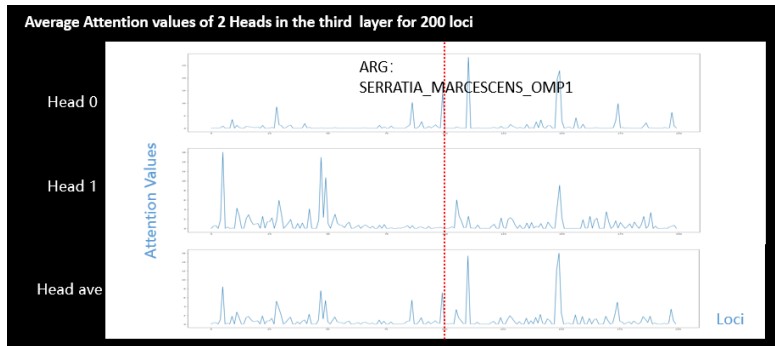

Figure 3: Visualization of attention values

### 3.3 GENOMICS INTERPRETATION

We used Attention Map based on attention mechanism to provide the interpretation of the model. We want to use this method to select a more interpretable set of features, and we will use biological knowledge to test its validity. To better assess the interpretability of the model, we performed statistics using known ARGs and loci. The table 2 shows the explanatory results of Attention Map which is in the style of Layers_Heads. The results show our methods could be in line with the knowledge of Genomics.

## 4 CONCLUSIONS

In this study, we used a very lightweight Transformer-based model (3 Encoder Block and 2 attention heads), and we got better prediction results than baseline. Our experiences shows that multi-task learning improved the generalization ability of the model, greatly reduced the training time of models for 14 tasks. Taking course learning as the training strategy of the model improves the accuracy and maybe reduces the influence of wrong labels. Our experience also shows that using Dynamic Task Prioritization can achieve competitive performance. Using learning progress signals to automatically adjust the weight of tasks is useful for muti-task learning. Our methods provide better biological interpretability than other methods (like k-mers). Our model got a more interpretable set of biological features.

Table 2: Results of explanatory results of Attention Map

| Antibiotic | Attention Map | ARG | |
|---|---|---|---|
| AMP | Layer3_head2 | OMPF | OmpF is the main route of membrane penet |
| AUG | Layer3_head2 | ACRB | Acrb is associated with crossresistance of c |
| AXO | Layer3_head2 | OMPF | OmpF is the main route of membrane penet |
| CHL | Layer2_head1 | SERRATIA_MARCESCENS_OMP1 | Loss or inhibition of OMP1 can prevent the |
| CIP | Layer3_head2 | MDSA | mdsA is associated with crossresistance of |
| FOX | Layer3_head2 | SERRATIA_MARCESCENS_OMP1, OMP36 | Loss or inhibition of OMP1 can prevent the |
| TIO | Layer3_head2 | OMPF | OmpF is the main route of membrane penet |

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
