# OpenReview forum: "Predicting Antimicrobial MICs for Nontyphoidal Salmonella Using Multitask Representations Learning "
_ICLR.cc/2023/Conference — Submitted to ICLR 2023_

### Official Review · Reviewer_uHZ8 · 2022-10-21

**Confidence:** 5
**Correctness:** 4
**Technical Novelty And Significance:** 1
**Empirical Novelty And Significance:** 1
**Recommendation:** 1

**Clarity, Quality, Novelty And Reproducibility:**

There are many grammatical issues, some of which are noted below. In particular, there is frequent errors with articles ("a" and "the").

P1: "The antimicrobial resistance (AMR) pathogens" > "Antimicrobial resistant pathogens".
P5: "Experience" > "Experimental".

**Strength And Weaknesses:**

The problem of predicting antimicrobial resistance is important and the authors' model seems reasonable.However, the methodological novelty is low and thus the primary result is the empirical performance.

Unfortunately, the empirical performance is impossible to judge because the authors compared only to a rule-based baseline. As the authors note, there have been many machine learning methods developed for predicting antimicrobial resistance. The authors must compare against these approaches.

Other comments:

As far as I can see, the "baseline" is not described anywhere. I assume that they used the same rule-based baseline as (Moradigaravand, 2018)

The authors cite a handful of AMR prediction methods but many are missing. For example, a Google Scholar search for "machine learning predict antimicrobial resistance" returns several reviews and papers not cited or compared against.



**Summary Of The Paper:**

The authors present an approach for predicting antimicrobial resistance in Salmonella based on genome sequence. The approach uses a transformer neural network trained in a multitask way. The authors found that this approach achieves better accuracy than a rule-based baseline.



**Summary Of The Review:**

See above.

---

### Official Review · Reviewer_cSGj · 2022-10-23

**Confidence:** 4
**Correctness:** 3
**Technical Novelty And Significance:** 1
**Empirical Novelty And Significance:** 1
**Recommendation:** 3

**Clarity, Quality, Novelty And Reproducibility:**

- The paper is straigthforward to read.
- The paper lacks any novel contribution.
- The paper lacks many implementation details and does not provide source code—making reproduction very challenging.


**Strength And Weaknesses:**

### Strength
- The paper is well written and easy to follow. The authors did a good job in explaining the importance of the application they are working on.
- Developing/improving data-driven predictors for healthcare application is aspirational and important.

### Weakness
- Although I think this is an important application, I feel that this paper lacks any machine learning novelty to fit this conference. The paper is based on a simple implementation of transformer and train the model with standard multi-task approach.
- The paper misses a lot of experimental results. It would be nice to (i) compare the model with other pre-existing approaches (rather than a simple “Baseline” method). It would also be good to see results that justify the proposed design choices (eg, ablation studies that explain why the model was design the way it was).
- Finally, the model does not provide any novel biological insight provided by the proposed model.

**Summary Of The Paper:**

In this papers, the authors propose a simple transformer-based model to predict minimum inhibitory concentration (MICs) . The trasnformer is built on the top of XGBoost-extracted features. The model is trained to simultaneously predict MIC of 14 antibiotics for Salmonela.

======== Post-rebuttal update ========

The authors did not provide a rebuttal. Therefore, I keep my rating.

**Summary Of The Review:**

Although I believe this is a very important application with huge potential impact, I believe this paper is not ready for publication in this conference. The paper lacks any machine learning contribution and the experimental results are not very convincing.

---

### Official Review · Reviewer_Ugwb · 2022-10-24

**Confidence:** 5
**Correctness:** 1
**Technical Novelty And Significance:** 1
**Empirical Novelty And Significance:** 1
**Recommendation:** 1

**Clarity, Quality, Novelty And Reproducibility:**

Clarity & Quality: The paper lacks much detail as mentioned in the weaknesses.

Novelty: There is zero methological novelty. The only contribution is that the paper applied Transformer for a niche task.

Reproducibility: The authors do not provide any source code, and the experimental details are lacking.

**Strength And Weaknesses:**

Strengths:
- According to the paper, this could be the first attempt to use Transformer with multi-task learning to predict the MIC of Salmonella for 14 antibiotics.
- Transformer does outperform an anonymous baseline, but the identity of the baseline is unknown.

Weaknesses:
- The paper makes zero methological contribution, which is the main interest for ICLR audience. The only contribution of this paper is applying Transformer for genome-based MIC prediction, which is better suited for biomedical venues (MLHC, ML4H, CHIL) or journals (JBI, JMIR Medical Informatics, BMC Genomics).
- The paper lacks much detail, especially regarding the experiment setup such as failing to describe the details of the baseline.

**Summary Of The Paper:**

This paper proposes to use Transformer combined with multi-task learning to predict the antimicrobial resistance, more specifically Minimum inhibitory concentration (MIC) of Salmonella against 14 antibiotics using its genome sequence. Experiment results show superior predictive performance compared to the baseline (although the paper does not specify which baseline), and visualization is possible thanks to the attention mechanism.

**Summary Of The Review:**

Overall, this paper offers zero methodological contribution, and even the empirical results are described with extremely lacking details.

---

### Decision · Program_Chairs · 2023-01-20

**Decision:**

Reject

**Justification For Why Not Higher Score:**

The reviewers have consistently commented on the improvement needed for this submission, such as on novelty of the method, more details for clarity. The authors didn't respond to these comments. Hence, this work would not merit acceptance in its current format.

**Justification For Why Not Lower Score:**

N/A

**Metareview: Summary, Strengths And Weaknesses:**

The authors describes a method to predict antimicrobial resistance in Salmonella strains from input genome sequences. They leveraged a transformer-based neural network with multi-task training. The authors showed that this method achieved better performance than a rule-based baseline.